# Flavones: Six Selected Flavones and Their Related Signaling Pathways That Induce Apoptosis in Cancer

**DOI:** 10.3390/ijms231810965

**Published:** 2022-09-19

**Authors:** Se Hyo Jeong, Hun Hwan Kim, Sang Eun Ha, Min Young Park, Pritam Bhagwan Bhosale, Abuyaseer Abusaliya, Kwang Il Park, Jeong Doo Heo, Hyun Wook Kim, Gon Sup Kim

**Affiliations:** 1Research Institute of Life Science and College of Veterinary Medicine, Gyeongsang National University, 501 Jinju-daero, Jinju 52828, Korea; 2Biological Resources Research Group, Gyeongnam Department of Environment Toxicology and Chemistry, Korea Institute of Toxicology, 17 Jegok-gil, Jinju 52834, Korea; 3Division of Animal Bioscience and Intergrated Biotechnology, Jinju 52725, Korea

**Keywords:** cancer, anticancer, apoptosis, flavonoids, flavones, cell signaling pathway

## Abstract

Cancer is a horrific disease that, to date, has no cure. It is caused by various factors and takes many lives. Apoptosis is a programmed cell death mechanism and if it does not function correctly in cancer cells, it can lead to severe disease. There are various signaling pathways for regulating apoptosis in cancer cells. Flavonoids are non-artificial natural bioactive compounds that are gaining attention as being capable of for inducing apoptosis in cancer cells. Among these, in this study, we focus on flavones. Flavones are a subclass of the numerous available flavonoids and possess several bioactive functions. Some of the most reported and well-known critical flavones, namely apigenin, acacetin, baicalein, luteolin, tangeretin, and wogonin, are discussed in depth in this review. Our main aim is to investigate the effects of the selected flavones on apoptosis and cell signaling pathways that contribute to death due to various types of cancers.

## 1. Introduction

### 1.1. Cancer and Apoptosis

Cancer is one of the world’s most deadly diseases, is expensive and complicated to treat, and has many side effects due to the chemical drugs used in its treatments. The causes of cancer are diverse and can be divided into two groups: intrinsic and extrinsic risk factors. Intrinsic risks arise from accidental errors in DNA replication and cannot be modified. Extrinsic risks include both endogenous and exogenous factors. Endogenous risk factors that can be partially modified include aging, hormones, growth factors, inflammation, and DNA repair mechanisms. Exogenous risk factors are adjustable and include radiation, chemical carcinogens, tumor-causing viruses, and smoking [1].

The hallmarks of cancer cells include the continuous maintenance of proliferation signals, avoidance of growth inhibitors, activation of invasion and metastasis, induction of angiogenesis, and the possibility of permanent proliferation. In addition, the abnormality of cellular energy metabolism, avoidance of immune destruction, tumor-promoting inflammation, genomic instability, and mutation have been suggested [2]. The malfunction of apoptosis is also a cause of cancer so cancer cells with apoptotic problems do not follow this mechanism [3].

Apoptosis, along with necrosis and autophagy, is a form of programmed cell death in cancers [4]. During the apoptosis mechanisms, there are significant morphological changes such as cell shrinkage, condensation of nuclear chromatin, DNA fragmentation, blebbing, and the formation of an apoptotic body [5,6]. Subsequently, phosphatidylserine released from the membranes causes phagocytosis and death from macrophages without an inflammatory response [7,8]. Characteristic highly regulated and programmed cell death without inducing the inflammation of apoptosis plays a crucial role in cancer prevention [9,10]. There are some crucial proteins for regulating apoptosis in cancer. First, cytochrome c (Cyt c) exists in the inner mitochondrial membrane. In the presence of dATP, Cyt c is released into the cytoplasm and combines with apoptotic protease activation factor-1 (APAF-1) and pro-caspase-9 to form apoptosome, causing cell death such as apoptosis [11,12]. The Bcl-2 family comprises members that promote, inhibit, and regulate apoptosis by controlling mitochondrial outer membrane permeability [11]. Bcl-2 families divide into the anti-apoptosis members Bcl-2, Bcl-xL, Bcl-w, and Mcl-1 and pro-apoptosis members BAX, BAK, and BID [13]. The tumor-suppressor protein p53 plays an essential role in the tumorigenesis and development of tumors [14]. When Fas combines with its ligand Fas L, it activates a lethal signal then ultimately causes a series of characteristic cellular changes and leads to apoptosis. During apoptosis in specific cells, the expression of Fas increases [15]. Survivin is another apoptosis-associated protein with a potent inhibitory role in apoptosis [16]. In addition, the caspase family of proteins, namely caspase-2, -8, -9, and -10, are upstream in the portion of the cascade and are the representative initiators of apoptosis. Caspase-3, -6, and -7 are the main factors that induce apoptosis and are downstream in the portion of the apoptotic cascade. In particular, caspase-3 is the main executioner in the apoptotic process [17]. Nuclear factor-kappa B (NF-кB) is a dimeric transcription factor protein complex that regulates cell inflammation, division, and apoptosis. The IкB kinase complex (IKK) is an upstream part of the NF-кB signaling cascade, which is one of the crucial regulators of cell survival, immune response, inflammation, and tumorigenesis [18]. Serine/threonine kinase protein kinase B, also called AKT, crucially regulates the balance between cell survival and death. The PI3K/AKT pathway is an intracellular signaling pathway that promotes metabolism, proliferation, cell survival, growth, and angiogenesis in response to extracellular signals. In addition, the PI3K/AKT pathway may regulate cell cycle progression and apoptosis [19,20]. 

In the course of cancer treatment, many studies have been conducted, and a sufficient understanding of apoptosis is considered to be important in dealing with cancer [3].

### 1.2. An Overview of Flavonoids and Their Structure–Activity Relationship in Anticancer Effects

To effectively prevent cancers through apoptosis and treat them without side effects, flavonoids as phenolic compounds play a significant role as natural biological response modifiers (BRM) [21]. Flavonoids are the secondary metabolites made by plants for self-protection from external factors and are the most common compounds present in plants and fungi such as vegetables and fruits. As shown in Figure 1, flavonoids are based on a 15-carbon skeleton consisting of two benzene rings (A and B rings), and then depending on the linkage through the heterocyclic pyran ring, flavonoids can be divided into six major subclasses, which include anthocyanidins, flavones, flavan-3-ols, flavanones, flavonols, and isoflavones [22,23,24]. Currently, more than 6000 flavonoids have been identified among the subclasses. There are many studies that show flavonoids induce apoptosis through various pathways and have anticancer effects [25,26]. Many researchers anticipate that these flavonoids will bring enormous benefits to the treatment of cancer [27].

As an anticancer effect, the structure–activity relationship revealed in the comparison of flavonoids showed that the adjacent di-OH at the 3′ and 4′ of the B ring and the C2-C3 double bonds were important in the potent antiproliferative activity [28,29]. In polymethoxylated flavones, C3-OH and C8 methoxyl groups play an important role in antiproliferation [30]. Compared to flavones, the hydroxyl group of C3 of flavonols does not show antiproliferative effects, and the high activity of myricetin compared to quercetin is explained by the presence of the third OH in the B ring adjacent to the catechol group [31,32]. In other studies, the di-OH 3′ and 4′, C2-C3 double bonds, and the carbonyl group C4 were shown to play important roles in the inhibition of proliferation through cell cycle arrest [33,34]. In addition, the structure–activity relationship of anticancer action through angiogenesis inhibition shows that flavanones are more effective at inhibiting the angiogenesis factor (VEGF) than flavones or flavanols. This is because the increase in the hydroxyl group of the B ring increases the inhibitory effect. Additionally, the absence of a double bond between C2 and C3 and the level of glycosylation may play an important role in angiogenesis inhibition [31,35,36].

### 1.3. An Overview of Flavones and Their Biological and Pharmacological Effects

Flavones are a subclass of flavonoids (Figure 1B) and are widely distributed in the plant kingdom including in parsley, celery, red pepper, and various kinds of herbs. The molecular formula of flavones is C_15_H_10_O_2_ and the chemical structure is C6-C3-C6. The basic flavonoid structure contains a ketone group at C4 and has a double bond between C2 and C3 [37,38]. Apigenin, acacetin, baicalin, luteolin, tangeretin, and wogonin are formed from the flavone core [39]. Researchers have found that flavones play a role in numerous biological activities such as protecting cell membranes [40,41], and can functions as antioxidants [42,43,44], xanthine oxidase inhibitors [45], and lipid-lowering agents [46]. Antiatherogenic agents [47], nitric oxide synthase inhibitors [48], cyclo-oxygenase inhibitors [49,50], leukotriene inhibitors [51], spasmolytic agents [52], NAD(P)H quinone acceptor oxidoreductase inhibitors [53], GABA antagonists [54,55], prostate hyperplasia therapeutics [56,57], antihyperglycemic [58], α-amylase inhibitors [59,60], glycogen phosphorylase inhibitors [61], aldose reductase inhibitors [62], sirtuin activators [63], hepatoprotective agents [41,64], antiarrhythmic agents [65,66], and anti-ulcer agents [67] are some biological functions of flavones. Flavones also function as antivirals [68], antibacterials [69], antifungals [70], antiprotozoals [71], photo-protectants [72], cosmetic agents [73], and phosphodiesterase inhibitors [74] (Table 1).

### 1.4. Flavones as Excellent Anticancer Agents

Flavones play a crucial role as anticancer agents [23,41] (Table 1). This is the reason that we focused on flavones and their anticancer relationship through apoptosis. In addition to the above six flavones, numerous flavones have biological and pharmacological effects on cancer through mitosis inhibition and angiogenesis inhibition. An abnormal cell cycle causes excessive cell proliferation, leading to cancer, and generating limitless new blood vessels is an advantage for cancer cell growth [41]. Another example of the biological and pharmacological effects of flavones on cancer is the inhibition of the protein tyrosine kinase. The binding of various growth factors to their receptors is related to cancer development. The abnormal activation of protein tyrosine kinase involved in this binding is a major target for anticancer treatments. Therefore, the inhibition of protein tyrosine kinase could be a useful anticancer strategy. For example, luteolin blocks the ATP binding site on the catalytic unit of protein kinase C (PKC). Chrysin also inhibits tyrosine kinase activity [41]. Flavone-8-acetic acid is involved in ornithine decarboxylase inhibition and the related anticancer activity. Rutin and luteolin are involved in aromatase inhibition related to breast cancer. Combretastatin acts as a powerful tubulin polymerization inhibitor, resulting in anticancer activity (Table 1) [41].

Overall, six flavones with anticancer effects were selected from numerous flavonoids, and the anticancer effects through the signaling pathways inducing apoptosis were investigated. In this paper, we review the apoptosis and signaling pathways that form the anticancer effects of flavones. This paper could therefore be useful for researchers studying the relationship between the anticancer effects of flavones and their signaling pathways that induce apoptosis.

## 2. Six Selected Flavones and Their Anticancer Effects

The anticancer effects of flavones have been studied by many researchers. Flavones showing typical anticancer effects include apigenin, acacetin, baicalein, luteolin, tangeretin, and wogonin. Figure 2 shows the chemical structures of flavones. These flavones are effective in treating breast cancer, lung cancer, stomach cancer, liver cancer, skin cancer, ovarian cancer, cervical cancer, and prostate cancer, all of which have high incidences and high mortality rates worldwide [23,75,76,77,78,79].

### 2.1. Apigenin

Apigenin is called 4′,5,7-trihydroxyflavone (Figure 2A). The molecular formula of apigenin is C_15_H_10_O_5_ and its molecular weight is 270.24 g/mol [80]. It is found in many plants, fruits, and beverages such as parsley, grapes, apples, chamomile tea, and red wine [80,81]. Many studies have revealed the anticancer effects of apigenin [82]. In particular, it is a flavone that is effective against breast, prostate, liver, skin, colorectal, and lung cancers [83].

### 2.2. Acacetin

Acacetin, also called 5,7-dihydroxy-4′-methoxyflavone (Figure 2B), has a molecular formula of C_16_H_12_O_5_ and a molecular weight of 284.26 g/mol [84]. Acacetin is mainly found in the safflower, propolis, and *Asteraceae* plants [85]. It is effective for treating breast, stomach, gastric, and prostate cancers [78,84,85,86,87].

### 2.3. Baicalein

Baicalein is mainly extracted from a plant called *Scutellaria baicalensis* and it is called 5,6,7-trihydroxyflavone [88,89]. Baicalein has a molecular formula of C_15_H_10_O_5_ and a molecular weight of 270.24 g/mol [89] (Figure 2C). It is effective against breast, liver, gastric, stomach, and ovarian cancers [88,89].

### 2.4. Luteolin

Luteolin, highly effective for a diverse range of cancers, is also called 3,4,5,7-tetrahydroxy flavone (Figure 2D). Its molecular formula is C_15_H_10_O_6_ and its molecular weight is 286.24 g/mol [90]. Luteolin is mainly found in abundance in broccoli, carrots, celery, cabbage, and parsley, and it is effective against breast, lung, stomach, liver, and cervical cancers [91,92,93].

### 2.5. Tangeretin

Tangeretin is called 4′,5,6,7,8-penta methoxyflavone (Figure 2E) and is mainly found in the citrus family. The molecular formula of this flavone is C_20_H_20_O_7_ and the molecular weight is 372.37 g/mol [94]. It is effective in treating breast, gastric, prostate, and bladder cancers [95,96,97].

### 2.6. Wogonin

Wogonin is called 5,7-dihydroxy-8-methoxyflavone (Figure 2F). The molecular formula of this flavone is C_16_H_12_O_5_ and its molecular weight is 284.26 g/mol [98,99]. Wogonin is mainly extracted from *Scutellaria baicalensis* Georgi (Lamiaceae) and is effective in treating breast, colorectal, lung, and ovarian cancers and glioma [98,100,101].

Table 2 shows the representative flavones that are effective against various cancers.

## 3. Apoptosis Pathways

In cancer cells, apoptosis is a very complex mechanism that is indispensable. If apoptosis does not function properly, cancer cells proliferate indefinitely [3]. As shown in Figure 3, there are two main pathways in apoptosis; one is the intrinsic pathway and the other is the extrinsic pathway [10]. 

### 3.1. Intrinsic Pathway

In the intrinsic pathway, cytochrome c is released from the mitochondria by the Bcl-2 family, which combines with apoptotic protease activation factor-1 (APAF-1) and ATP to form an apoptosome complex by binding pro-caspase-9. It activates caspase-9 and caspase-3, -6, and -7 which are the executioners of apoptosis [10,11,102]. 

### 3.2. Extrinsic Pathway

The extrinsic pathway is mediated by the death receptors including the Fas receptor, tumor necrosis factor receptor (TNF), and TNF-related apoptosis-inducing ligand receptor (TRAIL) [103]. These death receptors have death domains called the TNF receptor-associated death domain (TRADD) and the Fas-associated death domain (FADD) [3]. Death ligands (Fas L, TNF) bind to the death receptor to form a death-inducing signaling complex (DISC), which activates caspase-8 and caspase-3, the executioner of apoptosis [3,10,103,104].

Numerous signaling pathways are involved in these two major (extrinsic and intrinsic) apoptosis pathways. The regulation of apoptosis is imperative in cancer cells as well as some of the major cell signaling pathways involved in the survival and death of cancer cells.

## 4. Signaling Pathways Related to Cancer Cell Apoptosis

### 4.1. PI3K/AKT Pathway

The PI3K/AKT pathway, which can regulate apoptosis in cancer cells, plays a major role in mammalian cell proliferation, differentiation, autophagy, survival, and apoptosis [105]. The lipid kinase phosphoinositide 3-kinase (PI3K) consists of p110, a catalytic unit, and p85, a regulatory unit, which phosphorylates inositol carbon 3 of phosphatidylinositol 4,5-bisphosphate (PIP_2_) to phosphatidylinositol 3,4,5-triphosphate (PIP_3_) [106,107]. When PIP_3_ is increased, AKT and phosphoinositide-dependent kinase 1 (PDK1), which are enzymes in the pleckstrin homology domain (PH), are gathered and AKT is activated by PDK1 and mTOR complex 2 (mTORC2) [106,107]. This activated AKT phosphorylates BAD, caspase-9, and FOXO. Then, these proteins are inactivated, leading to the inhibition of apoptosis [108,109,110]. The activation and inhibition of the PI3K/AKT and mTOR pathways in human cancer cells determine the survival, carcinogenicity, metastasis, and invasion of cancer cells [111,112]. In cancer cells, this pathway is highly activated, and the inhibition of the PI3K/AKT and mTOR pathways induces apoptosis and has anticancer effects in various cancers [113].

### 4.2. Wnt Pathway

The Wnt pathway is a signaling pathway involved in cell proliferation, migration, stem cell differentiation, and various diseases including cancer. In this pathway, β-catenin belongs downstream of Wnt and plays a key role in apoptosis while triggering transcription by the Wnt ligand [114,115,116]. Wnt acts as a ligand in this pathway and its receptor is Frizzled [114]. In the absence of Wnt, destruction complexes such as Axin, glycogen synthase kinase-3 beta (GSK3β), adenomatous polyposis coli protein (APC), and casein kinase (CK) phosphorylate β-catenin. Due to proteasome protein degradation, β-catenin is degraded in the cytoplasm and cannot meet the T-cell factor (TCF) so transcription does not occur [114,117]. On the other hand, when the Wnt ligand binds to its receptor Frizzled, phosphorylation of the lipoprotein receptor-related protein (LRP), a co-receptor of Wnt, occurs, and Axin is phosphorylated and moves to the cell together with CK and GSK3β [114,115]. The β-catenin, freed from the influence of GSK3β, enters the nucleus and meets TCF to initiate β-catenin-dependent gene expression [114,117]. Inhibition of the Wnt/β-catenin signaling pathway showed anticancer effects by attenuating survival signals and inducing apoptosis [117,118]. Another mechanism is that β-catenin downregulates anti-apoptosis proteins and induces apoptosis by including transcription of p53 and c-myc, thus inducing the transcription of pro-apoptosis proteins [119].

### 4.3. JAK/STAT Pathway

Abnormal activation of the janus kinase (JAK)/signal transducer and activators of the transcription (STAT) pathway promotes tumorigenesis [120]. The cytokine receptors do not have tyrosine kinase activity so there is a problem in signal transduction. When cytokines, such as chemokines, interleukins, interferon, and TNF-α, a protein immunomodulatory agent secreted from immune cells, bind to a cytokine receptor without tyrosine kinase activity, JAK is gathered around the cytokine receptor to transmit a signal [121,122,123]. In this way, JAK phosphorylates STAT to form a STAT dimer and enters the nucleus to express the STAT target gene [123]. Blocking this JAK/STAT pathway in cancer cells suppresses the expression of the target gene that controls essential cellular function and prevents the avoidance of apoptosis, a characteristic of cancer cells, and an anticancer effect can be expected [124].

### 4.4. MAPK Pathway

The mitogen-activated protein kinase (MAPK) pathway plays an important role in cell proliferation, differentiation, angiogenesis, tumor metastasis, and apoptosis [125]. MAP kinase kinase kinase (MAPKKK) phosphorylates MAP kinase kinase (MAPKK) by external growth factors, stress, and cytokines. MAPKK phosphorylates MAPK, and MAPK phosphorylates and activates the target protein [125,126]. MAPKKK, MAPKK, and MAPK can be divided into four major signaling systems. The four-member classes in MAPK are extracellular signaling-regulated kinase1/2 (ERK1/2), ERK5, c-Jun NH-2-terminal kinase (JNK), and p38-MAPK. There are ERK1/2, ERK5, JNK1/2/3, and p38-MAPKα/β/γ/δ sub-families in each family [125,126,127]. Among them, ERK is known to have anti-apoptotic action, and JNK and p38-MAPK are likely to act in both anti- and pro-apoptosis [127]. An increase in ERK1/2/5 in cancer cells has been confirmed in the MAPK pathway, which could be a major target for anticancer effects [127,128,129]. The activation of JNK and p38-MAPK normally promotes but does not necessarily lead to apoptosis, and both JNK and p38-MAPK mediate anti-apoptosis and pro-apoptosis [130]. To explain apoptosis in the MAPK pathway, it can be anti- or pro-apoptosis depending on the response of the target protein to various stimuli or the molecular context according to the cell type [130].

### 4.5. p53

As a tumor suppressor, p53 is activated when DNA is damaged, causing cell cycle arrest and apoptosis. When this protein is mutated, cells that should die do not die and cancer occurs [131]. In the normal apoptosis process in the p53 pathway, when cellular stress signals occur, p53 induces the pro-apoptotic member BH-3-only proteins (BIM, NOXA, PUMA, and BAX) [132,133,134,135]. After this, BIM, NOXA, and PUMA suppress the Bcl-2 family of anti-apoptotic proteins, and apoptosis occurs through an intrinsic pathway by activated BAX and BAK [132,133,134].

Figure 4 shows the signaling pathways that are involved in apoptosis, which are interconnected. Numerous signaling pathways are involved in this apoptosis mechanism.

## 5. Regulation of Signaling Pathways in Which Six Selected Flavones Induce Apoptosis in Cancer Cells

### 5.1. Apigenin

Apigenin has exhibited anticancer effects on many cancer cells such as breast, prostate, liver, colorectal, skin, and lung cancers [83]. Apigenin induced apoptosis in cancer cells by downregulating various signaling pathways such as the PI3K/AKT pathway, ERK1/2, NF-кB, JAK/STAT, and Wnt/β-catenin [81]. First, in breast cancer, there was no induction of apoptosis through the intrinsic pathway because the mitochondrial membrane potential was not reduced in human epidermal growth factor receptor 2(HER-2)-expressing breast cancer BT-474 cells without the effect of Bcl-2 or BAX. However, there was an apoptotic process leading to the cleavage of caspase-8 and PARP via an extrinsic pathway. In addition, it has been demonstrated that breast cancer cells induce apoptosis through the inhibition of STAT3 signaling [136]. Apigenin also revealed that HER-2-expressing MCF-7 cells, another type of breast cancer cell, were involved in an apoptosis process that induced caspase-8 and PARP cleavage and apoptosis through a p53-dependent pathway [137]. In the case of prostate cancer, apigenin decreased Bcl-2 and Bcl-xL and increased BAX in PC-3 and DU145 human prostate cancer cells. In addition, the apoptosis inhibitors XIAP, c-IAP1, c-IAP2, and survivin were inhibited, resulting in apoptosis [138]. As a result of treatment with apigenin in Hep G2 cells, which are liver cancer cells, apoptosis was induced through the inhibition of the PI3K/AKT/mTOR pathway [139]. In HCT-116 colorectal cancer cells, apigenin increased tumor-suppressor proteins such as p53 and p21 [140]. In the case of SW480 cells, apoptosis was induced by increasing caspase-3 and BAX and decreasing Bcl-2 [141,142]. In addition, apigenin induced apoptosis by inhibiting the phosphorylation of STAT3 in colon cancer cells and simultaneously downregulating anti-apoptotic proteins such as Bcl-xL and Mcl-1 [143]. Apigenin induced apoptosis by regulating AKT and MAPK in A375SM human melanoma cells [144]. In A549 lung cancer cells, caspase-3, -8, and -9 were induced by apigenin and promoted cytochrome c, and apoptosis-inducing factor (AIF) was promoted through the mitochondria and caspase-3 and -9 were activated [145]. TRAIL-induced apoptosis by the upregulation of death receptors 4 and 5 occurred in p53-dependent NSCLC lung cancer cells. In addition, BAD and BAX were upregulated; Bcl-xL and Bcl-2 were downregulated; and NF-кB, AKT, and ERK activation were inhibited [146].

### 5.2. Acacetin

Acacetin induced apoptosis in cancer cells through various pathways such as mitochondria-mediated death signaling, caspase activation, the β-catenin pathway, and NF-кB/AKT signaling for breast, gastric, colorectal, and prostate cancers [78,84,85,86,87]. In breast cancer cells, MCF-7 and acacetin decreased Bcl-2 and released cytochrome c and AIF through the loss of mitochondrial membrane potential. Stress-activated protein kinase/c-Jun NH-4 terminal kinase1/2 (SAPK/JNK1/2) and c-Jun were activated and then apoptosis was induced through the SAPK/JNK1/2-c-Jun pathway [87]. In gastric cancer AGS cells, an apoptotic pathway was formed by caspase activity through ROS generation, mitochondrial mediation, and Fas activation [78]. In SW480 and HCT-116 colorectal cancer cells, the downregulation of the β-catenin pathway and increase in the mitochondrial membrane potential depolarization increased the BAX:Bcl-2 ratio and although there was no change in caspase, apoptosis was induced by AIF [86]. In DU145 prostate cancer cells, acacetin reduced phospho-AKT in a concentration-dependent manner, and phospho-GSK-3β, downstream of AKT, decreased and p53 increased. In addition, the activity of phospho-IкB and NF-кB was decreased, and apoptosis was induced through the NF-кB/AKT pathway due to the decrease in XIAP, and Bcl-2 [84].

### 5.3. Baicalein

Baicalein regulated apoptosis through various pathways for breast, liver, stomach, colorectal, lung, cervical, and ovarian cancers [88,89]. Baicalein regulated mitochondrial potential in MDA-MB-231 breast cancer cells, released cytochrome c, and activated caspase-3 to induce apoptosis [147]. In addition, increased p53 expression and increased ERK/p38 MAPK in MDA-MB-231 cells were associated with the pro-apoptotic effect of baicalein [89]. Additionally, in breast cancer cells MCF-7 and MDA-MB-231, Bcl-2 decreased and BAX increased. So, it induced apoptosis and autophagy through the inhibition of the PI3K/AKT pathway and downregulated the expression of phospho-AKT, phospho-mTOR, NF-кB, and phospho-IкB [147,148]. In HCC (hepatocellular carcinoma) liver cancer cells, baicalein increased BAX, decreased Bcl-2, and induced cleaved caspase-3, -9, and PARP. In addition, baicalein induced apoptosis by activating JNK [149]. In SGC-7901 gastric cancer cells, apoptosis was induced through the mitochondrial pathway. The mitochondrial membrane potential was disrupted so Bcl-2 decreased and BAX increased [150]. In HCT-116 and SW480 colon cancer cells, baicalein was involved in apoptosis by the MAPK/ERK and p38 pathways [88]. In A2780 ovarian cancer cells, baicalein decreased Bcl-2 and activated caspase-3 and -9 [151].

### 5.4. Luteolin

Luteolin induced apoptosis in breast, lung, gastric, liver, and cervical cancers [91,92,93]. First, in breast cancer, luteolin inhibited PI3K/AKT activation and increased FOXO3a activation, leading to cell cycle arrest and apoptosis [152]. Additionally, luteolin induced apoptosis through the downregulation of the human telomerase reverse transcriptase (hTERT) [153]. In SCLC lung cancer cells, luteolin caused cisplatin to become more susceptible to anticancer effects through JNK-activated apoptosis [154]. In NSCLC lung cancer cells, the expression of BAX, p53, p21, caspase-3, and -9 decreased and Bcl-2 increased due to microRNA-34a-5p inhibition. So, activation of microRNA-34a-5p blocked the caspase pathway and can be considered to cause apoptosis of lung cancer [155]. In BGC-823 gastric cancer cells, caspase-3, -9, and cytochrome c increased, the ratio of BAX to Bcl-2 increased, and apoptosis was induced through the inhibition of the MAPK and PI3K signaling pathways [156]. In SMMC-7721 liver cancer cells, luteolin increased caspase-8 and decreased Bcl-2 [75], and in Hep-G2 cells, it was reported that apoptosis was induced through BAX/BAK mitochondrial translocation and JNK activation [157]. In HeLa cervical cancer cells, luteolin increased the expression of various pro-apoptotic proteins, including BAX, BAD, BID, APAF1, TRADD, FAS, FADD, and caspase-3 and -9, but the expression of anti-apoptotic proteins, such as BCL-2 and MCL-1, decreased. Additionally, apoptosis occurred through the inhibition of the AKT and MAPK pathways due to the downregulation of MAPK, ERK1/2, and AKT and the upregulation of p53 at the transcriptional level [93].

### 5.5. Tangeretin

Tangeretin induced apoptosis in breast, gastric, prostate, and bladder cancers through various pathways [95,96,97]. The treatment of drug-resistant MDA-MB-231 breast cancer cells with tangeretin induced apoptosis by increasing BAX, caspase-3, and -8 and decreasing Bcl-*2* [158]. In AGS gastric cancer cells, caspase-3, -8, and -9, as well as BAX, tBID, and p53, were upregulated and apoptosis was induced through Fas/Fas L and the p53-dependent mitochondrial pathway [95]. In PC-3 and LNCaP prostate cancer cells, caspase-3 and cleaved caspase-3 and-9 were regulated. Only PC-3, BAX, and Bcl were regulated. Apoptosis was induced by inhibiting epithelial-mesenchymal transition (EMT), a cellular program important for cancer progression in these prostate cancer cells. In DU145 prostate cancer cells, apoptosis and DNA cleavage were induced through the regulation of the androgen receptor (AR)-PI3K/AKT/mTOR-Notch signaling pathway. In BFTC-905 bladder cancer cells, apoptosis was induced by regulating the release of cytochrome c and AIF from the mitochondria and cleaved caspase-3 and -9 and pro-caspase-3 and -9 [97].

### 5.6. Wogonin

Wogonin induced apoptosis through various pathways in breast cancer, colorectal cancer, lung cancer, ovarian cancer, and glioma [101]. Wogonin induced apoptosis by decreasing Bcl-2 and survivin and increasing BAX, p53, and caspase-3, -8, and -9 in MCF-7 breast cancer cells. In addition, apoptosis was induced through the inhibition of the PI3K/AKT/survivin signaling pathway and ERK activation [159]. In HT-29 colorectal cancer cells, Bcl-2 expression was reduced, BAX expression increased, and apoptosis was induced through the PI3K/AKT pathway [160]. In addition, in HCT116 cells, cell cycle arrest through the inhibition of the β-catenin-dependent Wnt signaling pathway showed attenuated proliferation and apoptosis in colorectal cancer [161]. In SW48, another colorectal cancer cell, apoptosis was induced by an increase in BAX and caspase-3, -8, and -9 and the inhibition of PI3K/AKT/STAT-3 [162]. In A549 lung cancer cells, wogonin induced apoptosis associated with increases in caspase-3, -8, and -9 and PARP and decreases in XIAP, c-FLIP, cIAP-1, -2, AKT, and STAT-3 [163]. In A2780 ovarian cancer cells, apoptosis was induced due to an increase in BAX and p53 and a decrease in AKT and Bcl-2 [98]. In U251 and U87 glioblastoma cells, wogonin induced not only the upregulation of caspase-3 and-9 but also PARP expression. In addition, it induced the generation of reactive oxygen species (ROS) and activated endoplasmic reticulum (ER) stress to induce apoptosis [101].

Comprehensively, Figure 5 shows the cell signaling regulation of apoptosis by apigenin, acacetin, baicalein, luteolin, tangeretin, and wogonin. Table 3 also shows the cancer cell lines, treatment concentrations, and apoptotic regulation of cell signaling pathways associated with the six above selected flavones.

## 6. Anticancer Effects of Six Selected Flavones in an In Vivo Model

Apigenin suppressed colorectal cancer and reduced tumor volume in prostate cancer without reducing appetite when ingested in Sprague–Dawley (SD) rats. Colorectal cancer was induced by azoxymethane (AOM), which is effective for the induction of colon carcinoma in SD rats. In these SD mice, apigenin treatment had the potential to suppress colorectal cancer by reducing aberrant crypt foci (ARF), which are clusters of abnormal tube-like glands, and increasing apoptosis [141]. In addition, in transgenic adenocarcinoma of mouse prostate (TRAMP) mice, apigenin inhibited the formation of prostate carcinoma by regulating the PI3K/AKT/FOXO pathway [164,165]. However, low bioavailability and instability in the gut were present [166,167]. Amounts of 20 and 50 μg of apigenin were administered to a xenograft model in which PC-3 prostate cancer cells were implanted into athymic nude mice. This resulted in a significant decrease in protein expression of XIAP and survivin in PC-3 tumor xenografts. A dose-dependent decrease in HDAC1 expression and an increase in Bax and PARP cleavage were observed in apigenin-treated mice [138].

Acacetin inhibited tumor growth through STAT3 regulation in DU145 prostate cancer cells in a nude mouse xenograft model [168].

As a result of analyzing the gene expression of a lung cancer xenograft tumor mouse model, the genes when baicalein induce apoptosis, ITGB3 (+6.96) and TNFRSF25 (+3.4), were most significantly upregulated [77]. In another study, mTOR inhibition prevented tumor growth in a breast cancer xenograft mouse model [169]. In addition, cervical cancer cell apoptosis was induced through an increase in BAX and a decrease in Bcl-2 in an in vivo tumor model [170].

Luteolin reduced the weight and volume of gastric tumors in a rat model by the inhibition of Notch1 and β-catenin [171]. When luteolin was administered to the lung cancer xenograft tumor mouse model, tumor suppression was shown by the inhibition of mutant EGF receptors, a major oncogene that induces tumorigenesis in many types of cancer, including NSCLC. This was followed by inducing the degradation of the EGF receptors and then preventing the PI3K/AKT/mTOR signal, which resulted in apoptosis and toxicity, and weight loss did not occur [172].

Tangeretin inhibited tumor growth in the MDA-MB-231 breast cancer cell nude mouse xenograft model [94]. Tangeretin had a strong inhibitory effect on tumor growth in the SGC-7901 gastric cancer cell nude mouse xenograft model [173].

As a result of treatment with wogonin in BALB/C nude mice xenografted with A2780 ovarian cancer cells and HT-29 colorectal cancer cells, the tumor volume and weight were reduced [174].

Table 4 shows the in vivo functions of anticancer effects in various cancer types.

## 7. Discussion and Conclusions

### 7.1. The Prospects of Compounds Such as Flavones and Flavonoids for Anticancer Effects

Cancer is still a disease whose treatments face many challenges. Anticancer research on polyphenols, such as flavonoids, has been actively conducted over the past few decades [175]. Of course, among flavonoids, flavones cannot be said to have the greatest anticancer effects and their mechanisms have not been fully elucidated. However, as shown in this review, flavones are effective in treating various cancers and they have special qualities that make them more effective than other flavonoids. This provides ample evidence to support flavones as having good prospects for treating cancer in the future so they are worth studying. However, in order to more clearly understand the effects of flavones and flavonoids, further studies are needed that focus on treatment effects, diet and bioavailability, absorption, metabolism, physiologically relevant models, and structural relationships between flavones and flavonoid subclasses [176]. In addition, it is important to proceed with clinical research using these data. There have been some clinical trials involving flavone-based agents, such as flavopiridol, which inhibits cdk and induces apoptosis, but more trials are needed [177].

### 7.2. Noteworthy ER-Mediated Apoptosis and the Need to Study Related Flavones

In this review, the apoptosis of the external and internal pathways has been mainly investigated, but the endoplasmic reticulum (ER)-mediated apoptosis pathway is also emerging as an important apoptosis pathway. The endoplasmic reticulum is an important organelle of eukaryotic cells and is necessary for maintaining intracellular homeostasis; controlling the synthesis, secretion, and folding of membrane-bound proteins; and transporting calcium ions [178]. When a misfolded protein accumulates in the ER and leads to activation of the unfolded protein response (UPR), it is called ER stress [178,179]. Through this ER stress, the combination with the mitochondria plays an important role in regulating apoptosis [178]. The first key protein in the ER stress-mediated apoptosis pathway is the C/EBP homologous protein (CHOP). The activation of the protein kinase R-like ER kinase (PERK) phosphorylates eukaryotic initiation factor 2 (EIF2a) and the activating transcription factor 4 (ATF4) is activated to upregulate the CHOP. The expression of this CHOP is also increased by activating transcription factor 6 (ATF6). The CHOP subsequently decreases the expression of anti-apoptotic Bcl-xL and Bcl-2 and conversely increases the pro-apoptotic BAX, BAK, NOXA, BIM, and PUMA [180,181]. Another pathway is that inositol-requiring enzyme 1 (IRE1) binds to tumor necrosis factor (TNF) receptor-associated factor 2 (TRAF2) to activate caspase 12, which in turn, activates caspase-3. In addition, the IRE1α-TRAF2 complex activates apoptosis signal-regulating kinase 1 (ASK1) and induces phosphorylation of JNK to induce apoptosis through c-jun, activates BAX and BAK, and inactivates Bcl-2 to play an important role in apoptosis [182].

Therefore, this endoplasmic reticulum-mediated apoptosis is very important and a pathway in which natural compounds, such as flavones, control ER stress [178] to act as an anticancer agent, which can be an important approach in the fight against cancer. Studies have been conducted on natural compounds, such as several flavonoids, but more research is needed.

### 7.3. Conclusions

This review includes a basic introduction to the selected flavones and their anticancer effects. In addition, we investigated the various pathways involved in apoptosis and discussed the signaling pathways by which flavones induce apoptosis. Flavones could be promising compounds for anticancer action in the future after further research considering this review information. Lastly, we hope this review will serve as a valuable manual for anticancer-related research using flavones.

## Figures and Tables

**Figure 1 ijms-23-10965-f001:**
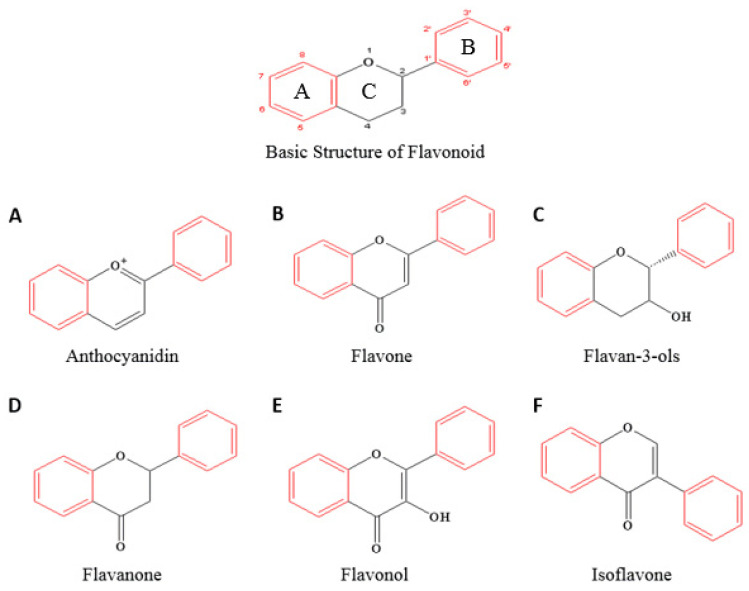
The basic structures and subclasses of flavonoids.

**Figure 2 ijms-23-10965-f002:**
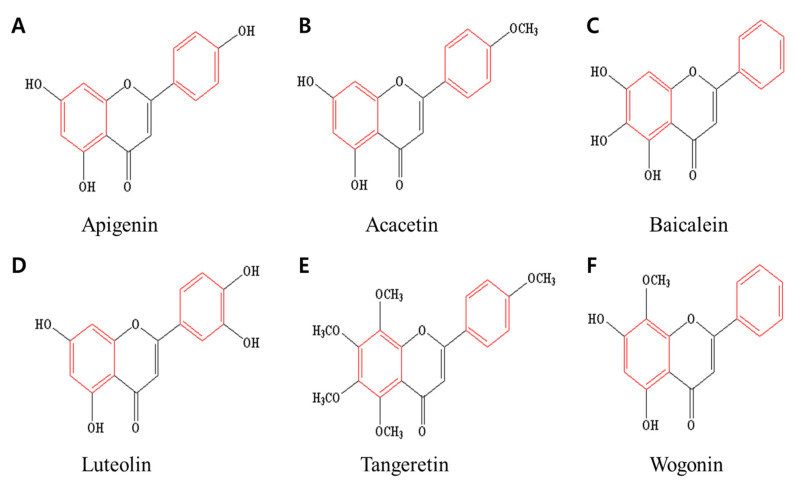
Chemical structures of flavones (**A**) apigenin, (**B**) acacetin, (**C**) baicalein, (**D**) luteolin, (**E**) tangeretin, and (**F**) wogonin.

**Figure 3 ijms-23-10965-f003:**
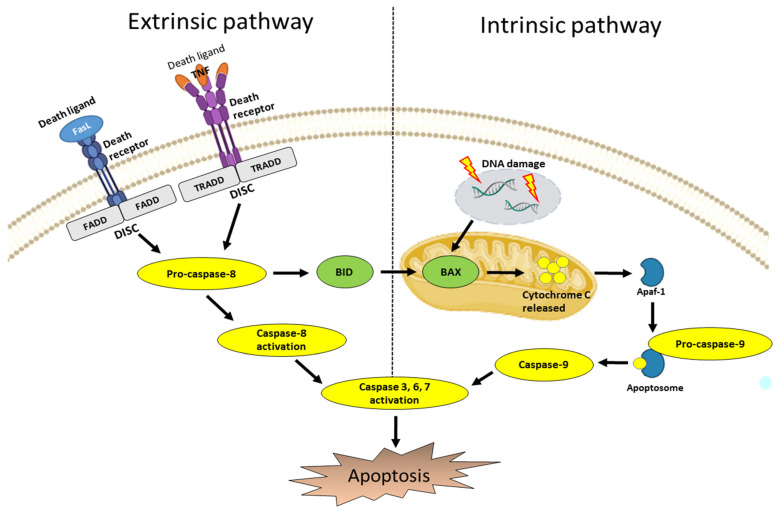
The extrinsic and intrinsic pathways of apoptosis.

**Figure 4 ijms-23-10965-f004:**
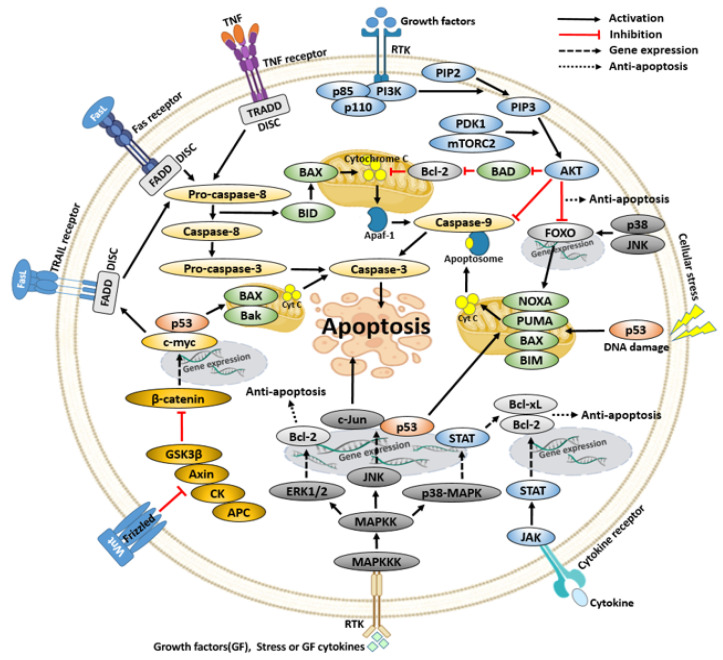
Various signaling pathways involved in apoptosis.

**Figure 5 ijms-23-10965-f005:**
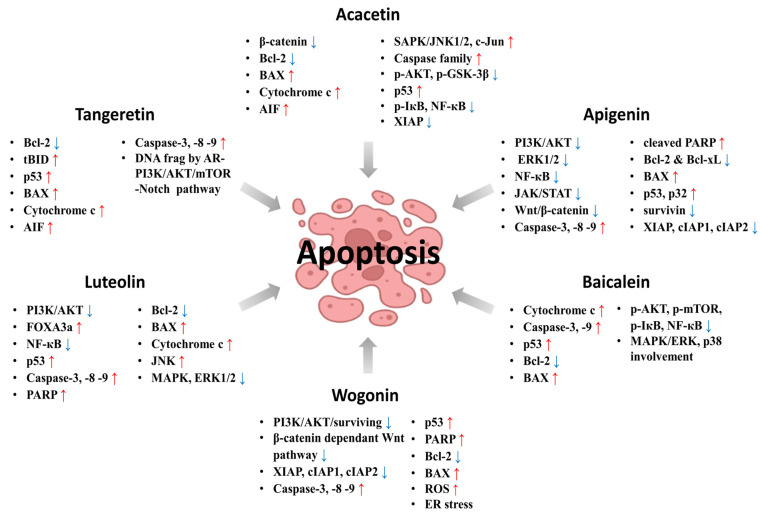
Up- and downregulation of related pathways and proteins in which cancer cells undergo apoptosis by flavones.

**Table 1 ijms-23-10965-t001:** Biological and pharmacological activities of flavones.

Biological and Pharmacological Activities of Flavones	Reference
Protecting cell membrane	[40,41]
Antioxidants	[42,43,44]
Xanthine oxidase inhibitors	[45]
Lipid-lowering agents	[46]
• Antiatherogenic agents - Nitric oxide synthase inhibitors - Cyclo-oxygenase inhibitors	[47,48,49,50]
Leukotriene inhibitors	[51]
Spasmolytic agents	[52]
Prostate hyperplasia therapeutics	[56,57]
Phosphodiesterase inhibitors	[74]
GABA antagonists	[54,55]
NAD(P)H quinone acceptor oxidoreductase inhibitors	[53]
Antiarrhythmic agents	[65,66]
Anti-ulcer agents	[67]
Antivirals	[68]
Antibacterials	[69]
Antifungals	[70]
• Antihyperglycemic - α-amylase inhibitors - Glycogen phosphorylase inhibitors - Aldose reductase inhibitors • Sirtuin activators	[58,59,60,61,62,63]
Antiprotozoals	[71]
Hepatoprotective agents	[41,64]
Photo-protectants	[72]
Cosmetic agents	[73]
• Anticancer agents - Mitosis inhibition - Angiogenesis inhibition - Protein tyrosine kinase inhibition - Ornithine decarboxylase inhibition - Aromatase inhibition • Tubulin polymerization inhibitor	[23,41]

**Table 2 ijms-23-10965-t002:** The six selected flavones with anticancer effects.

Flavone	Molecular Formula	Molecular Weight	Source	Anticancer Effects	Reference
Apigenin	C_15_H_10_O_5_	270.24 g/mol	Parsley, grapes, apples, chamomile tea, and red wine	Breast, prostate, liver, skin, colorectal, and lung cancers	[83]
Acacetin	C_16_H_12_O_5_	284.26 g/mol	Safflower, propolis, and *Asteraceae* plants	Breast, stomach, gastric, and prostate cancers	[78,84,85,86,87]
Baicalein	C_15_H_10_O_5_	270.24 g/mol	*Scutellaria baicalensis*	Breast, liver, gastric, stomach, and ovarian cancers	[88,89]
Luteolin	C_15_H_10_O_6_	286.24 g/mol	Broccoli, carrots, celery, cabbage, and parsley	Breast, lung, stomach, liver, and cervical cancers	[91,92,93]
Tangeretin	C_20_H_20_O_7_	372.37 g/mol	Citrus family	Breast, gastric, prostate, and bladder cancers	[95,96,97]
Woginin	C_16_H_12_O_5_	284.26 g/mol	*Scutellaria baicalensis* Georgi (Lamiaceae)	Breast, colorectal, lung, and ovarian cancers and glioma	[98,100,101]

**Table 3 ijms-23-10965-t003:** Relationship between flavones and signaling pathways leading to apoptosis in various cancer cells.

Apigenin				
Cancer	Cell Line	Treatment Concentration	Apoptotic Regulation of Cell Signaling Pathway	Reference
Breast Cancer	BT-474	20, 40, 60, 80, 100 μM	- Caspase-8 and PARP cleavage - STAT3 inhibition	[136]
	MCF-7	10, 20, 40 μM	- Caspase-8 and PARP cleavage - p53-dependent	[137]
Prostate Cancer	PC-3, DU145	5, 10, 20, 40 μM	- Bcl-2 and Bcl-xL decrease - BAX increase - Inhibition of XIAP, c-IAP1, c-IAP2, and survivin	[138]
Liver Cancer	Hep G2	10, 20, 40 μM	- PI3K/AKT/mTOR inhibition	[139]
Colorectal (Colon) Cancer	HCT-116	6.5, 12.5, 25, 50 μM	- p53 and p21 increase	[140]
	SW480	12.5, 25, 50, 100, 200 μM	- Bcl-2 decrease - Caspase-3 and BAX increase	[141,142]
	HT29, COLO320, DLD-1, HCT-116	5, 15, 50 μM	- STAT3 phosphorylation inhibition - Bcl-xL Mcl-1 decrease	[143]
Human Melanoma	A375SM	50, 100 μM	- AKT and MAPK regulation	[144]
Lung Cancer	A549	40, 80, 120, 160 μM	- Caspase-3, -8, and -9-inducing - Caspase-3 and -9 activation - Cytochrome c and AIF activation	[145]
	NSCLC	5, 10, 20, 40, 80, 160 μM	- TRAIL-inducing (Death Receptors 4 and 5-upregulated) - BAD and BAX increase - Bcl-2 and Bcl-xL decrease - NF-кB and ERK inhibition	[146]
**Acacetin**				
**Cancer**	**Cell Line**	**Treatment** **Concentration**	**Apoptotic Regulation of Cell Signaling Pathway**	**Reference**
Breast Cancer	MCF-7	25, 50, 100, 150, 200 μM	- Bcl-2 decrease - Cytochrome c and AIF activation - SAPK/JNK1/2 and c-Jun activation	[87]
Gastric Cancer	AGS	30, 60, 100 μM	- Caspase activation by ROS generation, mitochondrial-mediated, and Fas activation	[78]
Colorectal (Colon) Cancer	SW480, HCT-116	25 μM	- β-catenin downregulation - BAX increase, Bcl-2 decrease, and AIF-inducing	[86]
Prostate Cancer	DU145	12.5, 25 μM	- phospho-AKT decrease - phospho-GSK-3β decrease - p53 increase - XIAP and Bcl-2 decrease - Activity of phospho-IκB and NF-κB decrease	[84]
**Baicalein**				
**Cancer**	**Cell Line**	**Treatment** **Concentration**	**Apoptotic Regulation of Cell Signaling Pathway**	**Reference**
Breast Cancer	MDA-MB-231	25, 50, 75, 100 μM	- Cytochrome c release - Caspase-3 activation - p53 and ERK/p38MAPK increase	[89,147]
	MCF-7, MDA-MB-231	10, 20, 40 μM	- Bcl-2 decrease - BAX increase - PI3K/AKT inhibition - Downregulation of phospho-AKT, phospho-mTOR, NF-*к*B, and phospho-I*к*B	[147,148]
Liver Cancer	HCC	25, 50, 100, 200 μM	- BAX increase - Bcl-2 decrease - Cleaved caspase-3 and -9 and PARP-inducing - JNK activation	[149]
Gastric Cancer	SGC-7901	15, 30, 60 μM	- Mitochondrial-mediated - Bcl-2 decrease - BAX increase	[150]
Colorectal (Colon) Cancer	HCT-116, SW480	10, 20, 50 μM	- MAPK/ERK-mediated	[88]
Ovarian Cancer	A2780	20, 40, 80, 160 μM	- Bcl-2 decrease - Caspase-3 and -9 activation	[151]
**Luteolin**				
**Cancer**	**Cell Line**	**Treatment** **Concentration**	**Apoptotic Regulation of Cell Signaling Pathway**	**Reference**
Breast Cancer	Hs578T, MCF-7, MDA-MB-231	12.5, 25, 50, 100 μM	- PI3K/AKT inhibition - FOXO3a activation	[152]
	MDA-MB-231	10, 30 μM	- Downregulation of hTERT expression by NF-κB inhibitor α and c-Myc inhibition	[153]
Lung Cancer	SCLC	20 or 40 μM	- Increased anticancer effect of cisplatin through jnk activation	[154]
	NSCLC	10, 20, 40 μM	- Regulation of microRNA-34a-5p	[155]
Gastric Cancer	BGC-823	20, 40, 60 μM	- Caspase-3 and -9 increase - Cytochrome c increase - BAX increase - Bcl-2 decrease - Inhibition of MAPK and PI3K	[156]
Liver Cancer	SMMC-7721	25, 50, 100 μM	- Caspase-8 increase - Bcl-2 decrease	[75]
	Hep-G2	40, 80 μM	- BAX/BAK mitochondrial translocation - JNK activation	[157]
Cervical Cancer	HeLa	5, 10, 20 μM	- BAX, BAD BID, AFAF1, TRADD, FAS, and FADD increase - Caspase-3 and -9 increase - Bcl-2 and Mcl-1 decrease - Inhibition of AKT and MAPK	[93]
**Tangeretin**				
**Cancer**	**Cell Line**	**Treatment** **Concentration**	**Apoptotic Regulation of cell signaling pathway**	**Reference**
Breast Cancer	MDA-MB-231	4.5, 9, 18 μM	- BAX increase - Caspase-3 and-8 increase - Bcl-2 decrease	[158]
Gastric Cancer	AGS	10, 30, 60 μM	- Caspase-3, -8 and -9 increase - BAX, tBID, p53 increase - Fas/Fas L inducing	[95]
Prostate Cancer	PC-3, LNCaP	25, 50, 75, 100 μM	- Cleaved caspase-3 and -9 increase - EMT inhibition	[97]
	PC-3	25, 50, 100 μM	- BAX increase - Bcl family decrease	
	DU145	25, 50, 100 μM	- Regulation of the androgen receptor (AR)-PI3K/AKT/mTOR-Notch	
Bladder Cancer	BFTC-905	20, 40, 60 μM	- Cytochrome c and AIF release - Cleaved caspase-3, -9, pro-caspase-3 and -9 regulation	[97]
**Wogonin**				
**Cancer**	**Cell Line**	**Treatment** **Concentration**	**Apoptotic Regulation of Cell Signaling Pathway**	**Reference**
Breast Cancer	MCF-7	30, 60, 90 μM	- Caspase-3, -8, and -9 increase - BAX and p53 increase - Bcl-2 and survivin decrease - Inhibition of PI3K/AKT/survivin - ERK activation	[159]
Colorectal (Colon) Cancer	HT-29	25, 50, 100 μM	- Bcl-2 decrease - BAX increase - PI3K/AKT-mediated	[160]
	HCT116	10, 20, 40 μM	- Inhibition of the β-catenin-dependent Wnt	[161]
	SW480	4, 8, 16 μM	- BAX increase - Caspase-3, -8, and -9 increase - Inhibition of PI3K/AKT/STAT-3	[162]
Lung Cancer	A549	5, 10, 20 μM or 25, 30, 50 μM	- Caspase-3, -8, and -9 and PARP increase - XIAP, c-FLIP, cIAP-1, -2, AKT, and STAT-3 decrease	[163]
Ovarian Cancer	A2780	10, 20, 30 μM	- BAX and p53 increase - AKT and Bcl-2 decrease	[98]
Glioblastoma	U251, U87	4, 8, 16, 24 μM	- Caspase-3, -8, and -9 and PARP increase - Generation of ROS - Activation of ER stress	[101]

**Table 4 ijms-23-10965-t004:** Anticancer effects on flavones and in vivo mouse models.

Flavones	Type of Cancer	Mouse Model and Dosages	In Vivo Function	Reference
Apigenin	Colorectal cancer	AOM-injected SD rat	Suppressed colorectal cancer by reducing ACF and increasing apoptosis	[141]
	Prostate cancer	TRAMP mice (20 and 50 μg/mouse/day, gavage)	Inhibited the formation of prostate carcinoma by regulating the PI3K/AKT/FOXO pathway	[164,165]
		Athymic nude mouse (20 and 50 μg/mouse/day, oral)	Tumor volume was reduced; XIAP, survivin, and HDAC1 were downregulated; and BAX was increased	[138]
Acacetin	Prostate cancer	BALB/C nude mouse (50 mg/kg, intraperitoneal injection 5 days per week for 30 days)	Inhibited tumor growth through STAT3 regulation	[168]
Baicalein	Lung cancer	BALB/C nude mouse (1 and 3 mg/kg, intratumoral injection twice weekly)	Inhibition of tumor growth, genes that induced apoptosis, ITGB3 and TNFRSF25 upregulated	[77]
	Breast cancer	SCID-Beige mice (20 mg/kg, intraperitoneal injection for 5 days)	mTOR inhibition prevented tumor growth	[169]
Luteolin	Gastric cancer	BALB/C nude mouse (10 mg/kg, intraperitoneal injection 6 times)	Inhibition of Notch1 and β-catenin	[171]
	Lung cancer	BALB/C nude mouse (10 and 30 mg/kg, intraperitoneal injection daily for 15 days)	Tumor suppression and tumor weight were reduced by inhibition of PI3K/AKT/mTOR	[172]
Tangeretin	Breast cancer	Nude mouse (2.5 mg/kg, intraperitoneal injection once a week 4 times)	Inhibition of tumor growth	[94]
	Gastric cancer	BALB/C nude mouse (5, 25, 125 mg kg^−1^·bw^−1^·day^−1^)	Inhibition of tumor growth	[173]
Wogonin	Ovarian cancer	Athymic BALB/C nude mouse (20, 40, 80 mg/kg, intraperitoneal injection every 3 days)	Tumor volume and weight were reduced	[174]
	Colorectal cancer	Athymic BALB/C nude mouse (20, 40, 80 mg/kg, intraperitoneal injection every 3 days)	Tumor volume and weight were reduced	[174]

## Data Availability

Not applicable.

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
