# Peer review of "Flavones: Six Selected Flavones and Their Related Signaling Pathways That Induce Apoptosis in Cancer"

_ijms, 2022, doi:10.3390/ijms231810965_

Round 1
Reviewer 1 Report (Previous Reviewer 3)
I am satisfied with the changes the authors have made and the manuscript is now ready to be acceptable.
Author Response
For reviewer
Thank you for your attentive comments. and I would like to thank you for your positive evaluation, and I will work harder to achieve good research results.

Reviewer 2 Report (New Reviewer)
According to my opinion, the manuscript entitled "Flavone: Six Selected Flavones and Related Signaling Pathway that Induces Apoptosis in Cancer", given by Se Hyo Jeong, Hun Hwan Kim, Sang Eun Ha, Min Young Park, Pritam Bhagwan Bhosale, Abuyaseer
Abusaliya, Kwang Il Park, Jeong Doo Heo, Hyun Wook Kim and Gon Sup Kim merits to be published in International Journal of Molecular Sciences.
The Authors included in their review a lot of really important information and broad knowledge referring to flavones as agents inducing apoptosis in cancer.
However, chapter Discussion and Summary definitely has to be vaster due to the fact that in the current form it is neither a discussion nor a summary. Importantly, I see lack of deeper discussion referring to structure – activity relation. Moreover, perspectives issue, namely I wonder if flavones will remain important in the nearest future? Why not or why yes? To sum up, some revision of this manuscript is expected.
Author Response
For reviewer
Thank you for your attentive comments.
- Chapter Discussion and Summary definitely has to be vaster due to the fact that in the current form it is neither a discussion nor a summary. I see lack of deeper discussion referring to structure – activity relation.
Response: Thank you for your comment. I have corrected it. I added the structural activity relationship between flavonoids and some flavones in the introduction part. <1.2 An overview of flavonoids and their structural activity relationship on anticancer effect>. Also, as you commented, the discussion and conclusion parts have been refined into a more discussion format and summary format, and content has been added.
- Perspectives issue, namely I wonder if flavones will remain important in the nearest future? Why not or why yes?
Response: Thank you for your comment. I have corrected it. In the Discussion and Conclusion section, we added that flavones will still be important for the foreseeable future as we do not know the exact mechanism of flavones compared to their known potency. However, we have provided clear conditions that make this possible such as treatment effects, diet and bioavailability, absorption, metabolism, physiologically relevant models, and structural relationships.
Thanks you for your detail comment.

Reviewer 3 Report (New Reviewer)
Manuscript ID: ijms-1918312
Title: Flavone: Six Selected Flavones and Related Signaling Pathway that Induces Apoptosis in Cancer
Recommendation: The manuscript was well organized and easy to understand. Nevertheless, I think that some things should be solved.
Apoptosis is one hallmark of cancer cells; it is preferable that the authors refer to the rest of the cancer hallmarks in this review.
Recently, there is a promising interest in the role of the endoplasmic reticulum in the programmed death of cancer cells. We recommend the authors to consider endoplasmic reticulum-mediated apoptosis signaling pathway in this important review.
Author Response
For reviewer
Thank you for your attentive comments.
- Apoptosis is one hallmark of cancer cells; it is preferable that the authors refer to the rest of the cancer hallmarks in this review.
Response: Thank you for your comment. I have corrected it. I added additional cancer hallmarks including apoptosis below the first paragraph based on reviewer comments.
- Recently, there is a promising interest in the role of the endoplasmic reticulum in the programmed death of cancer cells. We recommend the authors to consider endoplasmic reticulum-mediated apoptosis signaling pathway in this important review.
Response: Thank you for your comment. I have corrected it. I added an ER-mediated apoptosis signaling pathway in the discussion section. Thanks to the good comments from the reviewer, it seems that more information about the ER-mediated apoptosis pathway has been learned and this review has enriched.
Thanks you for your detail comment.

This manuscript is a resubmission of an earlier submission. The following is a list of the peer review reports and author responses from that submission.
Round 1
Reviewer 1 Report
This review describes the role of flavones in apoptotic cell signaling. This topic is underexplored in the literature, so this review could fill an important gap. However, two major elements need to be addressed prior to publication.
First, the organization of the review makes it difficult to follow. A clearer outline, with an emphasis on the flavones, would greatly strengthen the manuscript. For example, in your intro goes into great detail about apoptotic mechanisms and does not mention flavones until much later. Other spaces, such as the description of tyrosine kinase signaling, are underrepresentated. A more clear outline that emphasizes flavones would make this review more impactful and novel.
Second, the English in this manuscript is difficult to read. Substantial editing is required to support publication of this review.
Author Response
This review describes the role of flavones in apoptotic cell signaling. This topic is underexplored in the literature, so this review could fill an important gap. However, two major elements need to be addressed prior to publication.
First, the organization of the review makes it difficult to follow. A clearer outline, with an emphasis on the flavones, would greatly strengthen the manuscript. For example, in your intro goes into great detail about apoptotic mechanisms and does not mention flavones until much later. Other spaces, such as the description of tyrosine kinase signaling, are underrepresentated. A more clear outline that emphasizes flavones would make this review more impactful and novel.
Second, the English in this manuscript is difficult to read. Substantial editing is required to support publication of this review.
- Thank you for your kind comment. I added flavones information at the end of the introduction and focused more on flavones. And I rearranged some sentences make my purpose clearer.
- Thank you for your kind comment. I have corrected some english problem.
Thanks again for your meaningful comment.
Reviewer 2 Report
The paper entitled - The Relationship between Flavone and Cell Signaling on Apoptosis in Cancer - discus an interesting subject describing the mechanisms by which the flavones can bi useful in cancer treatment.
The following observations has to be made:
Introduction
- please reformulate the sentence between line 40-42, 42-44, 46-47, 60-62, 84-86. all of these sentences are unclear.
- line 99-100 - replace "necessary manual" with "useful".
Table 1 - indicate the references for each biological and pharmacological activities of flavones
- line 111-112 - replace "parts" with "researchers".
- line 171 - replace "&" with "and".
The manuscript is incomplete - I suggest to add the experimental and clinical studies, already made and, eventually organize a table with these information in order to be clear and visible for the future research as you aimed with this paper.
Conclusions
-please reformulate sentence between lines 379-381 - is unclear

Author Response
The paper entitled - The Relationship between Flavone and Cell Signaling on Apoptosis in Cancer - discus an interesting subject describing the mechanisms by which the flavones can bi useful in cancer treatment.
The following observations has to be made:
Introduction
- please reformulate the sentence between line 40-42, 42-44, 46-47, 60-62, 84-86. all of these sentences are unclear.
Thank you for your comment. I have reformulated the sentence lines 40-42, 42-44, 46-47, 60-62, 84-86 and make it clear.
- line 99-100 - replace "necessary manual" with "useful".
Thank you for your comment. I have corrected it
Table 1 - indicate the references for each biological and pharmacological activities of flavones
Thank you for your comment. I have indicated the references for the table
- line 111-112 - replace "parts" with "researchers".
Thank you for your comment. I have corrected it
- line 171 - replace "&" with "and".
Thank you for your comment. I have corrected it
The manuscript is incomplete - I suggest to add the experimental and clinical studies, already made and, eventually organize a table with these information in order to be clear and visible for the future research as you aimed with this paper.
Thank you for your comment. For my future study about biochemical compounds such as flavonoid and flavone related to anticancer effects, I added the table about flavone and its apoptosis pathway in numerous cancer cells.
Conclusions
-please reformulate sentence between lines 379-381 - is unclear
Thank you for your comment. I have reformulated and changed the conclusion.
Thanks again for your kind and valuable comments.
Reviewer 3 Report
The paper by Kim and coworkers describes the summary of the anticancer effects of 6 flavones and their apoptosis pathways. The manuscript is well-organized and well-written. However, several points as indicated below need to be addressed by the authors before publication in the journal.
1. The format of author names should be corrected.
2. Line 19: Since flavonoids are chemical compounds, “non-chemical” should be removed.
3. Lines 22-23: Since the compound names are not proper nouns, they should not be capitalized.
4. Line 23: “this review” should be “in this review.”
5. Line 98: Since this manuscript is a review paper, “reviewed” should be used rather than “investigated.”
6. Lines 117-142: Please summarize these sentences as a table.
7. Lines 301-312 and 343-353: Please unitalicize these sentences.
8. Figure 4: The chemical structures of flavones should be removed, since they already appear in Figure 2.
6. Please check the format of figure legends, which include capitalized letters.
Author Response
The paper by Kim and coworkers describes the summary of the anticancer effects of 6 flavones and their apoptosis pathways. The manuscript is well-organized and well-written. However, several points as indicated below need to be addressed by the authors before publication in the journal.
- The format of author names should be corrected.
- Thank you for your comment. I have corrected author name format.
- Line 19: Since flavonoids are chemical compounds, “non-chemical” should be removed.
- Thank you for your comment. I removed “non-chemical” word.
- Lines 22-23: Since the compound names are not proper nouns, they should not be capitalized.
- Thank you for your comment. I have changed capitalized word to lower case.
- Line 23: “this review” should be “in this review.”
- Thank you for your comment. I have changed “this review” to “in this review.”
- Line 98: Since this manuscript is a review paper, “reviewed” should be used rather than “investigated.”
- Thank you for your comment. I have changed “investigated” to “reviewed”
- Lines 117-142: Please summarize these sentences as a table.
- Thank you for your comment. I have summarize to table.
- Lines 301-312 and 343-353: Please unitalicize these sentences.
- Thank you for your comment. I have unitalicized that sentences.
- Figure 4: The chemical structures of flavones should be removed, since they already appear in Figure 2.
- Thank you for your comment. I removed the chemical structure of flavones in figure 4
- Please check the format of figure legends, which include capitalized letters.
- Thank you for your comment. I have checked the format of figure legends and changed.
Reviewer 4 Report
The present manuscript can be recommended for publication. Still, it needs to be considerably revised due to technical problems and a lack of important information on the targeted subclass of secondary metabolites. In the revised manuscript, the following points should be addressed.
1. In the title, the authors may use selected flavones or similar expressions to reflect the content.
2. The authors should carefully review and include an additional section on structure-activity relationships and provide more representative flavones and derivatives reported in the literature, at least in the last two decades.
3. All authors should read the revised manuscript and correct grammatical and technical errors.
Author Response
The present manuscript can be recommended for publication. Still, it needs to be considerably revised due to technical problems and a lack of important information on the targeted subclass of secondary metabolites. In the revised manuscript, the following points should be addressed.
- In the title, the authors may use selected flavones or similar expressions to reflect the content.
- Thank you for your comment. I change my title from “Flavones: The Relationship between Flavone and Cell Signaling on Apoptosis in Cancer” to “Flavone: The Relationship between Six Selected Flavones and Cell Signaling on Apoptosis in Cancer.” And, thanks to your comments, this changed title will make this review clearer.
- The authors should carefully review and include an additional section on structure-activity relationships and provide more representative flavones and derivatives reported in the literature, at least in the last two decades.
- Thank you for your comment. This review paper focuses on the anticancer effect of the selected flavones by inducing apoptosis through a specific signaling pathway. It is a good idea to add a structure-activity relationship, but I think that the framework or direction of this review needs to be slightly changed to add the structure-activity relationships according to the flow of this review. However, next time, I plan to review flavones and their anticancer effects, focusing on the structurally active relationship. Thanks again for the nice comments.
As you can see from the reference year, the six flavones and flavone derivatives I selected were selected after investigating the flavones and flavone derivatives that have been actively researched within the last 20 years and are effective against notorious cancer. I thought it would be better if this content was added to the main text, so the following sentence was added to the second section, lines 133-135. -“These flavones are effective against breast, lung, gastric, liver, skin, ovarian, cervical, and prostate cancer, which are famous for having a high incidence and high mortality in worldwide and suffering many patients.”-
- All authors should read the revised manuscript and correct grammatical and technical errors.
- Thank you for your comment. we corrected grammatical and technical errors
Reviewer 5 Report
Thank you for sending me the research article paper “Flavones: The Relationship between Flavone and Cell Signaling on Apoptosis in Cancer” for review in the International Journal of Molecular Science the article of Se et al., the author discussed the role of flavone as a therapeutic agent for the cancer. Topic in interesting and novel. However, there are important points that should be improved.
1. Introduction part is too lengthy and needs to be divided into different subheadings.
2. Several review articles published on the therapeutic role of flavone against different types of cancers. What is the novelty of this review article?
2. Author should restructure the following paragraph “2. Flavones and their biological and pharmacological effects on cancer” and present in a scientific way. It would be easy for readers to understand the paragraph.
3. Author should discuss specific cancer types, rather than generally discuss the cancer. It looks too general.
4. Author should present the literature of cancer, signaling pathway, mechanism, flavone type treatment, experimental system, mice model in table form.
5. It would be great to present the clinical trial information of flavones against different cancer (tabular form).
6. Fig1 and 2 are too general. Author should present different flavone action and mode of action against specific cancer mechanisms.
Author Response
Thank you for sending me the research article paper “Flavones: The Relationship between Flavone and Cell Signaling on Apoptosis in Cancer” for review in the International Journal of Molecular Science the article of Se et al., the author discussed the role of flavone as a therapeutic agent for the cancer. Topic in interesting and novel. However, there are important points that should be improved.
- Introduction part is too lengthy and needs to be divided into different subheadings.
- Thank you for your comment. I divided into different subheadings.
- Several review articles published on the therapeutic role of flavone against different types of cancers. What is the novelty of this review article?
- Thank you for your comment. Many flavones and anticancer papers are usually focus separately on only apoptosis, and cell signaling. The novelty of this review is flavones and their anticancer effect through various cell signaling pathway not only apoptosis pathway but also apoptosis related signaling pathway. So, I focused on and put together flavones and cell signaling for cancer apoptosis. Focusing on flavone and choose 6 flavones (apigenin, acacetin, baicalein, luteolin, tangeretin, and wogonin) and choose major occurring cancer is also one of the characteristic of this review. As in many references in this paper, flavones were selected through literature information that among flavonoids, flavones are effective in major cancers.
- Author should restructure the following paragraph “2. Flavones and their biological and pharmacological effects on cancer” and present in a scientific way. It would be easy for readers to understand the paragraph.
- Thank you for your comment. I move the “biological and pharmacological effect of flavone” to introduction part. So paragraph 2 will be more scientific and clearer to understand.
- Author should discuss specific cancer types, rather than generally discuss the cancer. It looks too general.
- Thank you for your comment. The cancers primarily affected by flavones in this review are breast, lung, gastric, liver, skin, ovarian, cervical and prostate cancers. Sentence that these cancers are not ordinary cancers, but high-mortality and dangerous cancers, -These flavones are effective against breast, lung, gastric, liver, skin, ovarian, cervical and prostate cancer, which are famous for having a high incidence and high mortality rate in worldwide and suffering many patients.- are added with new reference. As a result, I think these sentences will be of great help in showing the effect of flavones by screening specific cancers rather than common cancers
- Author should present the literature of cancer, signaling pathway, mechanism, flavone type treatment, experimental system, mice model in table form.
- Thank you for your comment. Two new tables and parts for mouse models have also been added.
- It would be great to present the clinical trial information of flavones against different cancer (tabular form).
- Thank you for your comment. In discussion and conclusion part, I mentioned clinical trial but there are few clinical trials about my selected flavones and cancers.
(https://clinicaltrials.gov/ct2/results?cond=Cancer&term=flavonoid&cntry=&state=&city=&dist=).
- Fig1 and 2 are too general. Author should present different flavone action and mode of action against specific cancer mechanisms.
- Thank you for your comment. Fig 1 and 2 are simply inserted to explain the structures of flavonoids and flavones. I added the table 3 to shows the mechanism by which flavones act on specific cancer cells.
Round 2
Reviewer 1 Report
The revisions to this manuscript correct some minor grammatical errors and highlight the authors' intent in several places. However, the review's overall organization remains consistent with the previous version, which lacked clarity. More specific and actionable critiques are difficult given the lack of editing in writing. Substantial review by a native English speaker is strongly recommended.
Reviewer 2 Report
Even the authors the revised manuscript, the revised form do not meet scientific criteria to be published in this Journal.
I encourage the authors to reformulate their manuscript and to continue to improve it.
Reviewer 4 Report
The present manuscript needs to be improved and is not yet complete to be recommended for publication in its current form. The authors should provide an explicit limitation of this study. It would be more informative to briefly explain other potential flavone compounds with OH and OMe groups or their combination at different positions related to anticancer activity or a more detailed investigation of previously reported and the authors' opinions. (For example, apigenin and acetin are corresponding molecules with one or two OMe and so on).
Reviewer 5 Report
Author has fulfilled all items.